# Neonicotinoids impact bumblebee colony fitness in the field; a reanalysis of the UK's Food & Environment Research Agency 2012 experiment

Dave Goulson

School of Life Sciences, University of Sussex, Falmer, East Sussex, UK

## ABSTRACT

The causes of bee declines remain hotly debated, particularly the contribution of neonicotinoid insecticides. In 2013 the UK's Food & Environment Research Agency made public a study of the impacts of exposure of bumblebee colonies to neonicotinoids. The study concluded that there was no clear relationship between colony performance and pesticide exposure, and the study was subsequently cited by the UK government in a policy paper in support of their vote against a proposed moratorium on some uses of neonicotinoids. Here I present a simple re-analysis of this data set. It demonstrates that these data in fact do show a negative relationship between both colony growth and queen production and the levels of neonicotinoids in the food stores collected by the bees. Indeed, this is the first study describing substantial negative impacts of neonicotinoids on colony performance of any bee species with free-flying bees in a field realistic situation where pesticide exposure is provided only as part of normal farming practices. It strongly suggests that wild bumblebee colonies in farmland can be expected to be adversely affected by exposure to neonicotinoids.

## INTRODUCTION

Neonicotinoids are systemic neurotoxins, widely applied as seed dressings to arable crops, including those visited by bees, such as oilseed rape. When bees feed upon treated crops they are exposed to the pesticide at low concentrations in both the nectar and pollen (*Blacquiere et al., 2012*; *Goulson, 2013*; *Godfray et al., 2014*). A number of high profile studies have been published in recent years, suggesting a link between bumblebee health and exposure to neonicotinoid insecticides (e.g., *Laycock et al., 2012*; *Whitehorn et al., 2012*; *Gill, Ramos-Rodriguez & Raine, 2012*; *Feltham, Park & Goulson, 2014*; *Gill & Raine, 2014*). However, most of these studies can be criticised for not being representative of real, field situations since bees were forced to feed on pesticide-treated food, whereas in reality bees are free to choose where they forage. There is ongoing debate as to whether these studies accurately represent field exposure of bees to neonicotinoids (*Goulson, 2013*; *Godfray et al., 2014*). Nevertheless, on the basis of these studies and others on honeybees (*Henry et al.,*

Corresponding author
Dave Goulson,
D.Goulson@sussex.ac.uk

*2012*), the European Union recently voted to suspend use of the three most widely used neonicotinoids (imidacloprid, thiamethoxam and clothianidin) for use as seed dressings on flowering crops that attract bees, for a minimum of 2 years.

In 2013, the UK's Food & Environment Research Agency (FERA) published an online report describing the results of a field study they performed in 2012 on the impacts of neonicotinoid seed dressings on the development of bumblebee colonies (*FERA, 2013*). The study was specifically commissioned in response to the publication of *Whitehorn et al. (2012)*, which described an 85% drop in queen production in bumblebee colonies exposed for 2 weeks to field-realistic levels of imidacloprid. During the exposure phase of the Whitehorn study, the bees were confined and thus had no choice but to feed on treated food; the FERA study was an attempt to improve the realism of the experimental design by conducting this exposure phase with free-flying bees in the field.

The FERA study came to the conclusion that there was no significant link between pesticide exposure and colony performance, in contrast with the findings of *Whitehorn et al. (2012)*. However, attempts to improve the realism of the experiment by exposing bees in the field led to problems in standardising exposure between treatment groups, with neonicotinoids being recorded in control colonies. The European Food Standards Agency subsequently reviewed the study and raised concerns about how the authors had "elaborated and interpreted" their results (*European Food Safety Authority, 2013*). Here I present the results of a simplified re-analysis of the same data set based on the pesticide exposure of individual colonies.

## METHODS

In brief, the methods followed by FERA were as follows. Sixty young colonies of *Bombus terrestris audax* were obtained from a commercial supplier (Biobest, Belgium), and twenty of these colonies were each placed adjacent to one of three oilseed rape fields, treated with a neonicotinoid seed dressing of either clothianidin, imidacloprid, or neither (control). The colonies adjacent to the imidacloprid-treated field were placed out two weeks later than the other colonies, and were significantly smaller at the outset. Colonies were weighed and the number of workers counted at the outset of the experiment. Colonies were subsequently weighed weekly, and a single sample of stored nectar and pollen removed from each during the oilseed rape flowering period for neonicotinoid residue analysis. The oilseed rape fields had recently commenced flowering when the colonies were first placed out, and the colonies were removed after 6–7 weeks when flowering ceased. Colonies were then placed in sites with floral resources believed to be free of neonicotinoids. After a total of 8–9 weeks the colonies were senescing, and so were destructively sampled, at which point colony attributes such as numbers of new queens and numbers of workers were assessed.

Two sets of analyses were used. "Site based analyses" compared performance of the colonies at the three sites using Generalized Linear Models. "Residue based analyses" also used GLMs but analysed performance of the colonies with respect to the concentration of pesticide residues found in pollen or nectar samples, with site included as a fixed factor, and number of bees per colony at the outset included as a covariate. Separate

**Table 1 Percentage of 1,000 simulations producing a significant result for GLM's relating concentrations of pesticides in pollen and stored nectar to colony growth and queen production.** Values in brackets are after removal of two colonies with "high leverage."

|  | Thiamethoxam–pollen | Thiamethoxam–nectar | Clothianidin–nectar |
| --- | --- | --- | --- |
| Colony mass (mid-experiment) | 20.5 (2.3) | 0 | 35.9 |
| Colony mass—final | 90 (36.3) | 36.5 | 0.3 |
| Queen production | 74.8 (0) | 0 | 100 |

analyses were performed with regard to the concentration of clothianidin in nectar, and the concentration of thiamethoxam in either nectar or pollen. Thiamethoxam was found in the food stores of many of the colonies although it was not one of the experimental treatments. No imidacloprid was detected in any colonies, and no clothianidin in pollen. Residues of chemicals were often below the limit of detection (LOD), and simulated values between zero and the LOD were assigned randomly assuming a uniform distribution. This simulation was repeated 1,000 times and the proportion of runs where the effect of dose was found to be significant was reported. Where significant relationships were found, analyses were repeated after removal of colonies with "high leverage" i.e., colonies with high levels of pesticide residues.

## RESULTS AS REPORTED

The "Site based analyses" reveal significant differences between sites, with the colonies adjacent to the imidacloprid-treated field gaining much less weight than the others. These colonies also produced approximately 50% fewer new queens, though this difference was not statistically significant.

The results of the "Residue based analyses" are expressed as the proportion of 1,000 simulations which were statistically significant at $p < 0.05$. For example, in 90% of simulations there was a significant relationship between the concentration of thiamethoxam in pollen and the final weight of colonies, dropping to 36.3% when two "high leverage" colonies are removed (Table 1). For clothianidin in nectar, 100% of simulations were significant in explaining queen production i.e., there was always a significant, negative relationship between queen production and pesticide residue concentration.

However, when presented in the executive summary, the authors conclude that "No clear consistent relationships were observed" and "were neonicotinoids in pollen and nectar… to be a major source of field mortality and morbidity to bumblebee colonies, we would have expected to find a … clear relationship between observed neonicotinoid levels and measures of colony success." These conclusions are not in accordance with the results as described, although they do also state that there is a great need for further studies.

## A BRIEF CRITIQUE

The "Site based analyses" are not informative. Sites vary in numerous ways that will influence colony success, and there is no site-level replication in this study. The poor performance of the colonies next to imidacloprid-treated fields may simply be because these colonies were placed out later and were smaller to start with. The comparison

between the 'control' and either 'treated' field is confounded by the high levels of contamination of controls; indeed the concentrations of neonicotinoids found in the controls were greater than those in the 'imidacloprid' treatment.

The results of the "Residue based analysis" do not seem to be accurately represented in the summary or discussion. If there was no effect of pesticides then we would presumably expect the values in Table 1 to have a mean of 5 (due to type I errors where a null hypothesis is incorrectly rejected). Given that many of these values are substantially above 5, this would suggest a strong negative effect of pesticide residues on colony performance. For example, in the analysis of the effects of clothianidin, 100% of simulations detected a significant negative relationship. The removal of two colonies of "high leverage" in the analysis of the effects of thiamethoxam in pollen is not justified; these colonies do not appear to be outliers in the formal statistical sense, usually defined as point which falls more than 1.5 times the interquartile range above the third quartile or below the first quartile (Fig. 1A). The worth of randomly assigning values rather than simply assuming that all colonies in which no residue was detected have a value of zero, or the LOD, is not clear.

## A MORE SIMPLE ANALYSIS

Here, I present the results of simple Generalized Linear Models which, rather than assigning random values, simply either assume that (a) if no pesticide was detected, then none was present or (b) that if no pesticide was detected, it was present at the limit of detection. Otherwise, they are very similar to the original FERA "Residue based analysis," but because no data simulation was done, the results can simply be presented as significance tests in the traditional manner. My analyses used normal errors for colony weights, and negative binomial errors when numbers of queen produced was the response variable (data were overdispersed compared to a Poisson distribution). All analyses were performed in IBM SPSS Statistics 21. As in the original FERA analysis, 3 explanatory factors were included in each model: site (fixed factor); number of bees in each colony at the outset of the experiment (covariate); concentration of a particular pesticide in the nectar or pollen reserves within the nest (covariate). A separate analysis was conducted for each pesticide residue.

The concentration of clothianidin in nectar and of thiamethoxam in pollen negatively predicted both colony weight gain and queen production when using assumption (a) (Table 2). The negative effect of thiamethoxam in pollen on colony weight gain, and of clothianidin in nectar on queen production also remained significant under assumption (b) (Table 2).

It is clear that queen production is lower in colonies exposed to higher concentrations of neonicotinoids (Fig. 1). Amongst those colonies exposed to little or no pesticides there was considerable variation, with some colonies producing large numbers of queens and others few, but none of the colonies exposed to high levels of neonicotinoids produced large numbers of queens. Variation amongst the nests exposed to low concentrations is presumably due to other factors and is a common feature of bumblebee colonies
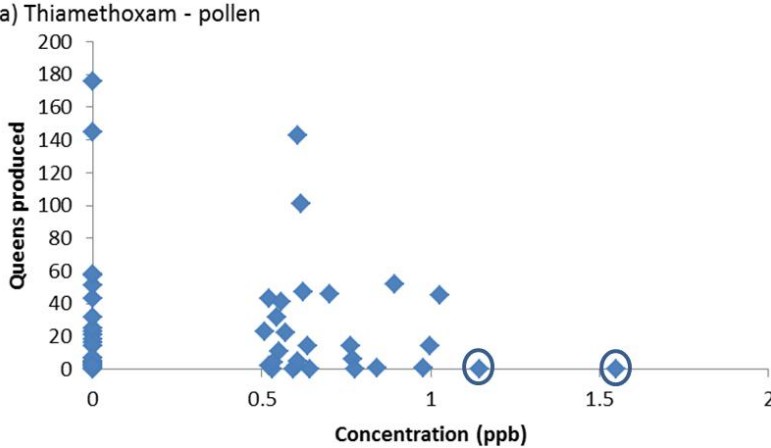

a) Thiamethoxam - pollen

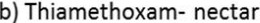

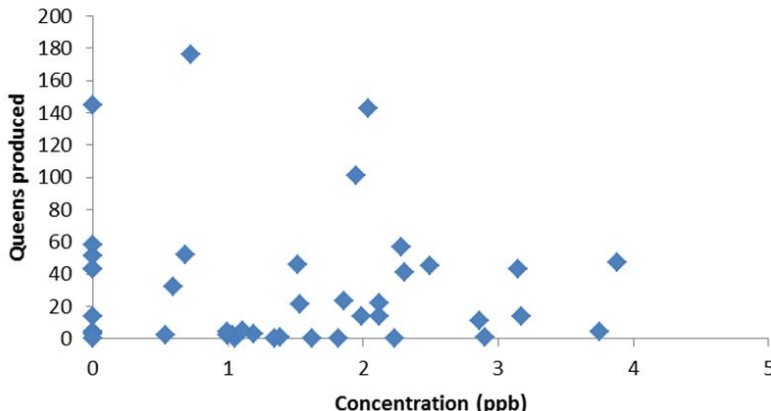

b) Thiamethoxam- nectar

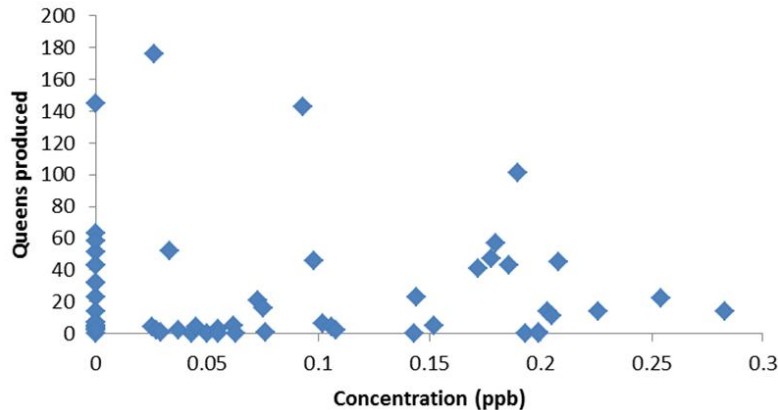

c) Clothianidin- nectar

**Figure 1  Queen production with respect to residues of pesticide in colony food stores ($n = 60$). Where no residue was detected a value of zero has been assigned.** The circles indicate two colonies that were removed from the FERA analysis. Figures are redrawn from *FERA (2013)*.

**Table 2 Effects of neonicotinoid concentrations in colony food stores on colony mass gain and queen production (assessed using individual GLMs for each pesticide, with negative binomial errors for queen number, and normal errors for colony weight).** The appropriateness of these distributions was verified during model checking. Site was included as a fixed factor, and pesticide concentration and the initial number of bees per colony as covariates. In all cases D.F. = 1. Where no pesticide was detected in a sample, we use one of two assumptions, either that there was no pesticide, or that the pesticide was present at the limit of detection. Both approaches give similar results.

| Pesticide concentration | Assuming < LOD = 0 | | | Assuming < LOD = LOD | | |
|---|---|---|---|---|---|---|
| | Parameter estimate | $\chi^2$ | $p$ | Parameter estimate | $\chi^2$ | $P$ |
| **Colony weight gain (g)** | | | | | | |
| Thiamethoxam in nectar | −109 | 3.35 | 0.067 | −115 | 3.13 | 0.077 |
| Clothianidin in nectar | −2476 | 9.48 | **0.002** | −1813 | 3.28 | 0.070 |
| Thiamethoxam in pollen | −223 | 4.92 | **0.027** | −445 | 6.40 | **0.011** |
| **Queen production** | | | | | | |
| Thiamethoxam in nectar | −0.367 | 1.68 | 0.195 | −0.304 | 0.881 | 0.348 |
| Clothianidin in nectar | −10.21 | 5.18 | **0.023** | −9.68 | 3.92 | **0.048** |
| Thiamethoxam in pollen | −1.24 | 4.93 | **0.026** | −1.83 | 3.19 | 0.074 |

(e.g., *Whitehorn et al., 2012*). Very similar plots are obtained if queen production is replaced by colony weight gain.

The full dataset is archived at Figshare: DOI 10.6084/m9.figshare.1320819.

# DISCUSSION

Despite the conclusions that were originally drawn by FERA, their data appear to provide the first clear evidence that colonies of free-flying bumblebees exposed to neonicotinoids used as part of normal farming practice suffer significant impacts in terms of reduced colony growth and queen production. The data also demonstrate that bumblebees in farmland are exposed to a cocktail of clothianidin and thiamethoxam in both nectar and pollen, since the large majority of colonies contained detectable quantities of both chemicals. It should be noted that clothianidin is a breakdown product of thiamethoxam, a well as a widely-used insecticide in its own right. Exposure frequently occurred even on the control farm where no neonicotinoids were used within 1 km of the bee colonies. This may be because neonicotinoid residues persist and can accumulate in soil from usage in previous years (*Goulson, 2013*), or because the bees travelled further afield than 1km to forage. Both seem likely, for *Bombus terrestris* are known to travel at least 1.5 km, and probably further (*Osborne et al., 2008*). Recent studies have detected neonicotinoid residues in wildflowers growing in field margins (*Stewart et al., 2014*), demonstrating that bees have multiple routes of exposure to neonicotinoids, not just when feeding on treated crops. This illustrates the extreme difficulty faced in trying to perform controlled experiments on the impacts of pesticides on free-flying bees.

The inescapable conclusion from these results is that wild bumblebee colonies located in hedgerows and woodland adjacent to or near arable farmland are likely to be experiencing

significant negative impacts on growth and queen production. Those colonies which, by chance, choose to feed upon flowers contaminated with higher levels of neonicotinoids are unlikely to produce many queens. This is very likely to have knock-on effects on population size the following year.

In December 2013 the EU commenced a two year moratorium on use of neonicotinoids on flowering crops. This study suggests that the moratorium ought to result in better performance of bumblebee colonies in farmland over time as neonicotinoid residues fall. It is unfortunate that no regular monitoring is taking place that might detect such benefits.

It is also concerning that the same data set can be interpreted in such wildly contrasting ways by different scientists, for this is likely to undermine confidence amongst the general public in the scientific process. The peer-review process, although not perfect, provides some degree of quality control; it would seem wise to encourage policy makers to base their decisions on research which has been through this process wherever possible.

## ACKNOWLEDGEMENTS

I am grateful to Helen Thompson for providing additional data on colony sizes at the beginning of the experiment.

### Funding

The author declares there was no funding for this work.

### Competing Interests

The author declares there are no competing interests.

### Author Contributions

- Dave Goulson analyzed the data, wrote the paper, prepared figures and/or tables, reviewed drafts of the paper.

### Data Deposition

The following information was supplied regarding the deposition of related data:
   Figshare
   http://dx.doi.org/10.6084/m9.figshare.1320819.

### Supplemental Information

Supplemental information for this article can be found online at http://dx.doi.org/10.7717/peerj.854#supplemental-information.

**Peer**J

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

## FURTHER READING

**DEFRA. 2013.** An assessment of key evidence about neonicotinoids and bees. Policy Paper PB13937.