# Peer review of "Neonicotinoids impact bumblebee colony fitness in the field; a reanalysis of the UK’s Food & Environment Research Agency 2012 experiment"

_PeerJ, doi:10.7717/peerj.854_

## Round 0.1 · original submission · Minor Revisions

This MS is nicely written and clearly conveys its main points. It is a reanalysis of existing data and it serves to potentially provide more clarity on the topic of neonicotinoids and pollinators.

While both reviewers call for minor revisions, those revisions are, in my mind, *vital* to ensuring that these results can be adequately scrutinized.

In particular, both reviewers – either directly or indirectly – state that since this is a statistical paper there needs to be a better discussion of the statistical methods.

For instance:

"It would have been nice if more explanation was given to the major differences between the disparate authors’ statistical approaches, and their appropriateness." (Reviewer 1)

and

"...there should be some more clarity on the output of the analysis and considering aspects of the data." (Reviewer 2)

Yes, it is a fairly simple GLM, etc. But, as Reviewer 2 points out, the author should "...provide the statistical software used, version and perhaps the scripts used for statistical analysis for others to confirm."

Both reviewers also provide a number of other points relating to the analysis. Again, since this paper is all about the analysis, the author needs to make sure that readers can precisely replicate the process on their own with the same data.

On the topic of the data, the author should ensure that the data in the analysis are fully and permanently available in citable format (i.e. with a DOI). That may mean asking for permission to post the data in a DOI-based repository such as figshare, Dryad, or some other such service. The DOI for the data file should then be referenced in this paper.

Both reviewers made other substantive (as well as more minor) comments, and I encourage the author, of course, to consider them all carefully.

Both reviewers wish to remain anonymous. However the author does have the option of publishing the anonymous reviews alongside the eventual paper following a successful revision and rebuttal process. I would strongly encourage this to ensure full transparency, as this is a topic of understandable concern and importance.

Reviewer 1 ·

Basic reporting

• The submission is not a research article per se, but rather a reanalysis of data reported previously by different authors. I am not familiar enough with PeerJ policies regarding the suitability of such a study to make a clear recommendation regarding this point.
• The article is clearly written, including enough introduction and background information for readers. However, it would have been informative to have been given more information about previous critiques of the report, namely, what were the main concerns of the European Food Standards Agency. In what way did they feel the authors ‘elaborated’ and ‘interpreted’ their results? Did the author’s concerns echo that of the Agency?
• The article conforms to PeerJ required templates.
• Figure 1 of the article is an amalgamation of several figures from the original study; figure 1a = figure 7B(a), figure 1b = figure 7C(a), and figure 1c = figure 7A(a). Figure 1c is referenced on line 126, prior to the first mention of figure 1c on line 127. Following the original order, figure 1c should be moved to become figure 1a. I do not recall any mention by the authors of where their figures originated. They should make it clear that their figures are reproductions from the original authors’ publication. Furthermore, the authors combined data from the three sites, which were given different symbols in the original study. The authors should remind readers in the figure legend that they have combined data for the three sites.
• The submission is self contained, representing an appropriate ‘unit of publication’.

Experimental design

• The submission does not describe original primary research (see above).
• The submission does however provide a question. The authors investigated whether removing ‘site’ as an additive factor in analyses (the original authors’ data analyses included a ‘site’ factor), would clear-up the ambiguity reported in the original publication of the effect of neonicitinoids on bee health.
• The investigation included changing how the original analyses were conducted, removing site as an additive factor (above), and changing how pesticide residues below the level of detection (LOD) were used in analyses. The authors also kept two ‘outliers’ in analyses the original authors did (and did not) remove.
• The authors reanalyze the data without ‘site’ as an additive factor in analyses, but leaving other parameters similar to the authors’ original analyses. The authors analyzed the original data replacing bootstrapped values with either the LOD values or 0 values. Did the original authors provide means for pesticide residue values for individual bootstrap runs? If so, how different might the mean be from either the LOD or 0? In other words, what is the main difference between the original authors’ and the current author’s treatment of the data? What’s the advantage, if any?

Validity of the findings

• From their exercise, the authors conclude that the original study, and indeed the original analyses (or at least parts presented in the original study) clearly show the negative effect of exposure of neonicitinoids on bee health, contradicting the conclusions presented in the executive summary of the original study. The author restates portions of that summary that support their claim that the study was biased against presenting results showing significant negative effects of neonicitinoids on bee health. However, the author excluded other portions of the summary that state that the “study was not a formal test of the hypothesis”, and pointing to “the great need of further studies under natural conditions”. These phrases do not give total support that neonicitinoids do not negatively impact bees, but rather that the study was inconclusive.
• I do appreciate that the authors have demonstrated that neonicitinoids could, in fact, negatively impact bee health in natural conditions, even when exposed to very low doses of these pesticides. A simple commentary on the original publication would not have been sufficient to point to the significant results ‘buried’ within it. An exercise, like the current submission, was likely necessary to point to these hidden results. It would have been nice if more explanation was given to the major differences between the disparate authors’ statistical approaches, and their appropriateness.

Additional comments

No Comments.

Reviewer 2 ·

Basic reporting

Clear and well written

Experimental design

No Comment

Validity of the findings

I feel more information is required on the statistics.

Additional comments

This MS focusses on a re-analysis of the data from the DEFRA report “Effects of neonicotinoid seed treatments on bumblebee colonies under field conditions”. The author argues that (i) the original data analysis used in the report was inappropriate and that (ii) the presentation of the results, as presented in the report, did not accurately represent the data.

The FERA report has previously been criticised since its publication in 2013 including, but not limited to, the experimental design, lack of standardisation between treatments, and the presence of neonicotinoids in control colonies. However, despite the obvious flaws, the study has been influential and was used to oppose a limited moratorium on neonicotinoid use in the EU. Given the importance attached to it by policy makers in the UK government, any analysis (such as this one) that assesses the validity of this study is of general interest and suitable for publication.

Overall, the MS is clear and well written and most of my comments are minor (see below). However, I do have one concern that I think should be addressed before publication, and I raise it simply because of the importance of getting it right due to its potential implication.

Major Comment

I should first highlight that I am not a statistician and nor do I have any extensive expertise in statistical analyses, but given the paper entirely hinges on the statistical analysis performed here I think there should be some more clarity on the output of the analysis and considering aspects of the data. The method describing the analysis seems very sensible, but:

 It has not been discussed about the dispersion of the data. From eyeing the data in the figures I think there is a good chance you will have over-dispersion which may require quasi-poisson error distributions. If there is over-dispersion this can significantly affect the p-values.
 In the text the author states that they provide the full model outputs, yet I don’t think is true. I am not sure what the parameter estimate truly represents, and no co-efficients or intercepts have been stated.
 Your parameter estimate for Clothianidin is massively higher than the other two. Why is this? Does this suggest over-dispersion?
 In your figures you provide what seems to be a fitted a linear regression line. The figures represent count data so this is not really correct, and it is also a bit misleading as given you have used concentration as a covariate in your model you should be able to plot the line as part of your statistical output. I think you would then find a curved line which would likely be quite skewed in appearance.
 General: Given the paper is a critique; could the author provide the statistical software used, version and perhaps the scripts used for statistical analysis for others to confirm?


Minor Comments:

66-67: “rendering comparisons between field treatments difficult” This seems more like a point to bring up in the discussion.

118-122: I feel this section is a little unclear. I am unsure why the values in Table 1 should have a mean value of 5. To my mind, the example given in 122-128 does not clarify why we should expect a value of 5 with no effect – could the author clarify with a more explicit example?

139: This may be personal taste but I don’t like statements about p values being close to or not quite significant. If the experimental design lacked statistical power and so only very large effects could be detected explaining why large differences in means were not giving a p-value below 0.05 – then this should be stated rather than a statement of being close to significance.

142: “assumption a or b” neither assumption is labelled a or b. Delete “a or b” as not required.

143: “triangular distribution” - would this be easier to understand by rephrasing it as a negative relationship between queen number and pesticide concentration.

---

## Round 0.2 · accepted · Accept

The author has responded adequately to all of the reviewers' concerns, and so I recommend publication of this MS.

The author states, in response to Reviewer 2:

"I would be happy to provide full model outputs in supplementary online materials."

I would strongly encourage publication of the full model outputs.